# Privacy-Preserving Broker-ABE Scheme for Multiple Cloud-Assisted Cyber Physical Systems

**DOI:** 10.3390/s19245463

**Published:** 2019-12-11

**Authors:** Po-Wen Chi, Ming-Hung Wang

**Affiliations:** 1Department of Computer Science and Information Engineering, National Taiwan Normal University, Taipei 11677, Taiwan; 2Department of Information Engineering and Computer Science, Feng Chia University, Taichung 40724, Taiwan; mhwang@fcu.edu.tw

**Keywords:** attribute-based encryption, multiple cloud-assisted cyber–physical system, deniable encryption

## Abstract

Cloud-assisted cyber–physical systems (CCPSs) integrate the physical space with cloud computing. To do so, sensors on the field collect real-life data and forward it to clouds for further data analysis and decision-making. Since multiple services may be accessed at the same time, sensor data should be forwarded to different cloud service providers (CSPs). In this scenario, attribute-based encryption (ABE) is an appropriate technique for securing data communication between sensors and clouds. Each cloud has its own attributes and a broker can determine which cloud is authorized to access data by the requirements set at the time of encryption. In this paper, we propose a privacy-preserving broker-ABE scheme for multiple CCPSs (MCCPS). The ABE separates the policy embedding job from the ABE task. To ease the computational burden of the sensors, this scheme leaves the policy embedding task to the broker, which is generally more powerful than the sensors. Moreover, the proposed scheme provides a way for CSPs to protect data privacy from outside coercion.

## 1. Introduction

Cyber–physical systems (CPSs), first proposed in Rajkumar et al. [1], is a concept in which field data are monitored and collected by sensor systems and are then relayed to computer systems. Computer systems then use computer-based algorithms to make decisions regarding how field agents should proceed to obtain better results and pass these commands on field agents. This architecture is also the main concept underpinning Industry 4.0 [2,3] and the internet of things (IoT) [4].

Today, cloud computing provides convenient, reliable, and on-demand services and is becoming increasingly popular. Accordingly, the cyber component in CPS is being extended to cloud computing to conform to the trend of the time. In 2013, National Institute of Standards and Technology (NIST) integrated cloud computing and CPS and proposed an architecture known as the cloud-assisted cyber–physical system (CCPS) [5] (actually, NIST uses the term cyber–physical cloud computing (CPCC). In this paper, we prefer to use CCPS since this term is consistent with the term CPS). The associated white paper [5] lists five benefits of CCPS: (1) efficient use of resources, (2) modular composition, (3) rapid development, (4) smart adaption to the environment at every scale, (5) reliability and resiliency.

The modular composition aspect is very important. Since it is almost impossible for one cloud service vendor to provide every type of service, many cloud services are composed of multiple sub-cloud systems. For example, when a user wants to choose a restaurant, he may draw upon a number of cloud systems simultaneously to do so, including a map service, a customer comment service, and a booking service. This is also the case in CCPS. Field sensors collect different types of data. Each CSP may require a specific subset of data for analysis, decision-making, and the provision of its own unique service. For example, in a hospital, there may be lots of sensors in a patient, and they periodically report to different health-care systems, like the disease tracking system, patient caring system, or even air conditioning system. This facility is called a multiple CCPS (MCCPS). For this scenario, a new entity called a broker is introduced between the field sensors and CSPs [6,7]. The broker is responsible for dispatching sensor data to appropriate CSPs according to their requirements or CSP properties. In general, the IoT gateway can play the broker role. Compared to the sensor, the IoT gateway is usually considered to be much more powerful. So it is reasonable to move most computational works from the sensor to the broker. Figure 1 shows a diagram of the broker architecture for the MCCPS.

To secure data communication in the MCCPS, attribute-based encryption (ABE) is an appropriate technique since sensor data are required by multiple CSPs at the same time. By embedding the appropriate policy-checking mechanism, only CSPs that satisfy pre-defined policies are authorized to access field data. As such, data can be encrypted once and then be forwarded to multiple possible cloud service vendors. However, most ABE schemes require intensive computational resources. Considering the limited sensor capabilities, applying existing ABE schemes to sensors may not be a good idea.

Data privacy is another important issue in the MCCPS. Since data are processed in clouds, some entities may force clouds to release field data through some type of power enforcement. For example, a patient may wear many biosensors to monitor his/her health status. These biosensors forward collected data to the cloud for further analysis and health determination. If some third party forces the cloud service vendor to release the patient’s health information, traditional encryption schemes would not protect user privacy since CSPs handle unencrypted data. Other than expecting the CSP to reject outside requests directly, CSPs must be provided with a tool to prevent coercion and ensure data privacy.

To solve these two problems, we propose a privacy-preserving broker-based ABE scheme in which the policy generation role is moved from the sender, which is the field sensor in this scenario, to the broker. The reason for this is that the broker is typically more powerful than the sensor. As for the data privacy issue, we use a deniable encryption property to protect CSPs from being coerced. When being coerced by outside forces, CSPs can release fake data that include persuasive evidence, and the outside forces cannot reject claims from the CSPs since the claims are indistinguishable from those of real data. Thus, the real data are kept safe from coercion.

A deniable encryption scheme is a kind of non-commitment encryption scheme since it is possible to interpret an existing ciphertext into a fake message other than the user data. Some deniable encryption schemes will be described in Section 2. In general, these schemes cost a lot of resources and it is not affordable for a sensor to implement them. For example, Canetti et al. proposed a bitwise deniable encryption scheme that requires each transmission bit has an independent encryption environment [8]. To overcome this problem, our design is pre-determined and multi-distributional. We show that the cost of our scheme is reasonable for a sensor in Section 4.

### 1.1. Contributions

In this paper, we present a deniable broker-based CP-ABE scheme for the MCCPS scenario. The advantages of this scheme are as follows:Broker-based ABE. Most ABE schemes consider only the sender and receiver. In a ciphertext-policy ABE (CP-ABE) scheme, the sender must embed a policy-checking mechanism in the ciphertext generation process. Generally speaking, the computational cost of this process depends on the number of attributes included. When more attributes are taken into account, more computation power is required in the encryption process. That is, when applying CP-ABE in the MCCPS scenario, many attributes are required and the field sensor may not have sufficient resources to encrypt the collected data.

To solve this problem, in this scheme, the broker assumes the policy embedding task. The sensor encrypts the collected data without considering the required policy. The semi-finished ciphertext, which as yet remains confidential, is forwarded to the broker. The broker then processes the received ciphertext and re-encrypts the ciphertext into a new version which requires policy checking upon decryption. As such, the encryption step, which requires the most computational power, is moved from the sensor to the broker. Therefore, broker-based ABE is more suitable for the MCCPS scenario.
Arbitrary policy support without dedicated keys. The broker-based ABE is a variant of the proxy-ABE, where the broker role is similar to that of a proxy. This intermediary step transforms the given ciphertext into a new ciphertext. The proxy-ABE requires a re-encryption key, which is often bundled with the new policy. However, this is not practical in the MCCPS scenario since there are many types of sensors, many CSPs, and having a re-encrypt key for each combination would be almost impossible. Besides, as described above, the sensor cannot complete the policy embedding task due to its limited computational resources. So, the broker does not transform the ciphertext from one policy into another, but rather adds a policy-checking step to an existing ciphertext. Moreover, without transformation by the broker, CSPs cannot decrypt these messages from the sensors. This ensures that a message from a field sensor must pass through the broker and have had the policy-checking step appended before being received by the CSP. Thirdly, although the broker is an intermediate entity that participates in the encryption process, it is not allowed to see the data from field sensors. This arrangement is known as a semi-trusted broker.

In this work, to achieve this feature, we use a composite order group, which can be divided into two subgroups. The sender, which is a sensor in this scenario, encrypts a message in the composite order group. The broker and the CSP have keys for different respective subgroups. When receiving a ciphertext, the broker first removes the secret key for the particular subgroup. Then, the broker applies the attribute encryption mechanism to the other subgroup and forward the new ciphertext to the CSP. Note that no re-encryption key is required when embedding policies. The broker only takes public information to embed the access structure into the ciphertext. Since the CSP has the key of the subgroup, the CSP can correctly decipher the message.
Lightweight and blockwise deniable ABE. Deniable encryption is an encryption scheme in which the sender and receiver can persuade others that a given ciphertext is from a fake rather than a real message. Most deniable encryption schemes are bitwise encryption schemes that encrypt only one bit at a time. Undoubtedly, a bitwise encryption can support any kind of message by repeating the encryption process many times. However, in practice, this is not efficient especially, when considering the MCCPS scenario, since the sensor cannot support this approach due to its resource limitations.

Following the concept proposed by Chi et al. [9], in this work, we constructed a blockwise deniable encryption scheme for the MCCPS scenario. The sensor can easily generate two messages, one using real data and the other fake data, and prepare two convincing pieces of evidence respectively. The evidence is taken from a chameleon hash function, for which the collision derivation is very lightweight, so the overall computation required by the sensor is much less than that required in other deniable encryption schemes.

Note that in this MCCPS scenario, we focus only on the sensor and CSP in providing the deniability feature, without considering the broker. The reason for this is that the broker cannot see the content of the ciphertext, and can, therefore, make no claims about the content.
Multiple brokers support. In the MCCPS scenario, there are many cloud services that may be shared by sensors from different fields. That is, one cloud service must simultaneously support multiple brokers and the sensors that are under these brokers. It is a trivial matter that each broker and CSP share one unique encryption environment, which is usually the public key. However, this will increase the management complexity since many brokers and multiple public keys are required.

In this work, we separate public information into two parts. The first part comprises system-wise public information and the second broker-wise public information. Broker-wise public information is only shared by one broker and the sensors that are under this particular broker. The CSP needs only system-wise public information. This is an advantage that will decrease the public key management burden on the CSP.

### 1.2. Organization

The rest of the paper is organized as follows. In Section 2, we briefly introduce some existing ABE, deniable encryption (DE) techniques and the preliminaries used in this scheme. We present the proposed scheme is in Section 3. In Section 4, we evaluate the scheme’s security and performance. In Section 5, we further discuss the proposed scheme. We then conclude the paper and address future work in Section 6.

## 2. Related Works

### 2.1. Previous Works on Attribute-Based Encryption

ABE is an encryption system in which only those who satisfy some specific conditions can successfully decrypt the ciphered data. This ABE concept was first introduced by Sahai and Waters [10] (although identity-based encryption (IBE) is also a type of ABE and was first proposed in Boneh et al. [11], here we only focus on ABE review and do not consider IBE schemes). There are two types of ABE; key-policy ABE (KP-ABE) and cipher-policy ABE (CP-ABE). Their difference lies in where the policy-checking step occurs. KP-ABE embeds the decryption policy in the key generation process, whereas CP-ABE embeds the policy in the encryption process. Goyal et al. proposed the first KP-ABE construction [12] and Bethencourt et al. proposed the first CP-ABE [13]. Both these schemes support monotonic formula policy checking. In this paper, we focus only on CP-ABE schemes because CP-ABE enables the sender to decide who is allowed to see the content. The MCCPS scenario implies that the broker can decide which cloud services should be launched and that the policy can be modified by the broker. As such, we only review CP-ABE related techniques. Waters proposed the first fully expressive CP-ABE based on linear secret sharing schemes (LSSS) [14]. In this paper, we use Water’s CP-ABE as the base scheme and enhance it to a privacy-preserving broker-ABE. Lewko et al. [15] improved the Waters scheme [14] to a fully secure CP-ABE, albeit with some efficiency loss. Attrapadung et al. [16] constructed a CP-ABE that supports a constant-size ciphertext. Tysowski et al. [17] designed their CP-ABE scheme to support dynamic key management for mobile users.

There is a variant ABE called proxy-ABE, which is a concept for importing a new entity called a proxy that can re-encrypt ciphertexts. When the sender encrypts data for some group, the proxy can transform the ciphertext from one receiver group to another. In the re-encryption process, the proxy is not allowed to decrypt the ciphertext. In this paper, since the broker is at the same logical location as the proxy and also performs the re-encryption process, we review some proxy-ABE studies. Luo et al. [18] first proposed ciphertext-policy attribute-based proxy re-encryption (CP-AB-PRE), which allows a proxy to change the receiver group from one to another. However, their scheme only supports the AND-gate operation. Li [19] enhanced the Luo et al. scheme [18] by using a matrix access structure. Liang et al. built a secure proxy-ABE in the chosen-ciphertext model [20]. Chandar et al. [21] used proxy to re-encrypt ciphertexts for removing the access right of revoked users. Touati et al. [22] studied the issue of private key updating in a CP-AB-PRE scheme.

Some researchers also focus on how to apply ABE on the IoT environment. Yao et al. claimed that the bilinear operation is expensive, so they built their scheme the elliptic curve cryptography without pairing [23]. Oualha et al. made the sensor compute lots of elements in advance to accelerate the encryption procedure [24]. However, this approach required the sensor to have a large space or to interact with a trusted party. Some used fog/edge nodes as the intermediate layer and implement mechanisms on them. Jiang et al. focused on the key-delegation abuse problem and solved this problem with a traitor [25]. Zuo et al. developed ABE with outsourced decryption for the fog computing [26]. Li et al. outsourced the exponential operation to speed up both the encryption and decryption process [27]. Fischer et al. used a proxy to convert a ciphertext into an ABE ciphertext [28]. The problem of Fischer approach is that it is an interactive model. The receiver needs to communicate with some authority before decryption.

In this paper, we propose an ABE scheme called broker-based ABE. We outsource partial encryption works to the entity called broker. Although the broker ABE is a kind of proxy-ABE, it is actually more than that. The broker-ABE is an idea that integrates secret-sharing and proxy re-encryption, whereby the sender encrypts a message using the secret shared by the broker and CSP. So, neither the broker nor the CSP can decrypt the ciphertext individually. For re-encryption, the broker can add a policy-checking mechanism to ciphertexts to verify the CSPs’ attributes.

### 2.2. Previous Works on Deniable Encryption

The idea of DE was first proposed by Canetti et al. [8]. Like other encryption schemes, DE provides semantic security to protect encrypted data. In addition, DE allows the sender and/or receiver to persuade outside coercers that the given ciphertexts have been encrypted from fake messages. A number of DE techniques have been proposed. Since ABE is a public key encryption system, here we only review some deniable public-key encryption works. In the scheme proposed by Canetti et al. [8], the authors used a translucent set to provide fake messages with convincing evidence. A translucent set is one that contains a trapdoor subset. It is easy to choose a random element from the universal set or the subset, but it is difficult to determine if a given element belongs to the subset without the use of the trapdoor. If a sender wants to encrypt one bit 0, the sender sends an element not contained in the subset. To encrypt one bit 1, the sender sends an element that is contained in the subset. When being coerced, the sender simply claims a bit from 1 to 0 by claiming the random element from the universal set that lies in the subset coincidentally. Although this is a sender-deniable scheme, Canetti et al. proved that this scheme can be extended to a bi-deniable scheme through an interactive model. Following this idea, Durmuth et al. [29] used samplable encryption to build a translucent set. O’Neill et al. [30] constructed a bi-translucent set based on a lattice. So, without using an interactive model, this scheme can provide both sender and receiver deniability, which is called bi-deniability.

Apart from translucent-based schemes, other techniques are used to build DE schemes. O’Neill et al. [30] made use of a simulatable public key system and used the voting approach to provide deniability. O’Neill et al. also proposed a new concept called multi-distributional deniability. They proposed two sets of algorithms and their outputs are computationally indistinguishable. So a user can use one set but claims using the other one. This approach can support non-interactivity and fully deniability at the same time. Gasti et al. [31] proposed a DE scheme that creates a secret key pair between the sender and receiver. Chi et al. [9] proposed a DE scheme based on composite order groups and used chameleon hash functions to make fake data convincing.

In our proposed scheme, we use a DE scheme to protect user data privacy in CSPs. When being forced to release field data, CSPs can provide false data to outside coercers, with evidence in the existing ciphertexts. Note that there are three types of DE schemes: sender-deniable encryption, receiver-deniable encryption, and bi-deniable encryption. The names specify the entity that can generate fake evidence. In the MCCPS scenario, since senders are usually sensors with limited resources and cannot store data for very long, this paper focuses on a receiver-DE scheme to protect the data privacy of data stored in CSPs. The broker is not allowed to see the content of ciphertexts and therefore cannot participate in the coercion process.

### 2.3. Preliminaries

#### 2.3.1. Prime Order Bilinear Groups

Let G and $G_{T}$ be two multiplicative cyclic groups of prime order *p*, with map function $e:G\times G\to G_{T}$. Let *g* be a generator of G. G is a bilinear map group if G and *e* have the following properties:Bilinearity: ∀u,v∈G and a,b∈Z, $e\left(u^{a},v^{b}\right)=e{\left(u,v\right)}^{ab}$.Non-degeneracy: e(g,g)≠1.Computability: the group action in G and map function *e* can be computed efficiently.

#### 2.3.2. Waters CP-ABE Scheme

In this paper, we extend the Waters CP-ABE scheme [14] to a privacy-preserving broker CP-ABE scheme. Waters used the LSSS as the basis of his construction. To form the secret, he treated attributes as secret shares and policies. So if an entity can rebuild the secret, this implies that the entity has enough attributes to satisfy the given policies. The LSSS used by Waters is as follows:

**Definition** **1**(LSSS: Linear Secret Sharing Schemes [32])**.**
*A secret sharing scheme *Π* over a set of parties P is called linear (over $Z_{p}$) if*
*1.* The shares for each party form a vector over $Z_{p}$.*2.* There exists an l×n matrix M called the share-generating matrix for *Π*. For all i=1,…,l, the i’th row of M is labeled by party ρ(i), where ρ is a mapping function from {1,…,l} to party field P. When considering column vector $v=(s,r_{2},\ldots ,r_{n})$, where $s\in Z_{p}$ is the secret to be shared and $r_{2},\ldots ,r_{n}\in Z_{p}$ are randomly chosen, Mv is the vector of l shares of secret s according to *Π*. The share ${\left(Mv\right)}_{i}$ belongs to party ρ(i).

Based on the above definition, an LSSS has a linear reconstruction property. That is, with an LSSS Π, an access structure A, and valid shares of a secret *s*, *s* can be easily recovered. Beimel [32] showed that the recovery procedure is time polynomial to the message size. For an ABE scheme, a party in P can be treated as an attribute. So if the receiver has attributes that satisfy the decryption policy, this implies that the receiver can recover the secret *s* and correctly decrypt the message. The Waters CP-ABE scheme includes the following algorithms:Setup()→(MSK,PK): This algorithm chooses a bilinear group of prime order *p* with generator *g*, random elements $\alpha ,a\in Z_{p}$, and hash function $H:{\left\{0,1\right\}}^{*}\to G$. The public key PK is $\left\{g,e{\left(g,g\right)}^{\alpha },g^{a}\right\}$ and the system secret key MSK is $g^{\alpha }$.Encrypt(PK,(M,ρ),M)→CT: Given message *M* and LSSS access structure (M,ρ), this algorithm first chooses a random vector $\vec{v}=\left(s,y_{2},\ldots ,y_{n}\right)\in Z_{p}^{n}$. Let M be an l×n matrix and $M_{i}$ denote the *i*th row of M. This algorithm calculates $\lambda_{i}=\vec{v}M_{i},\forall i\in \left\{1,\ldots ,l\right\}$. Next, this algorithm chooses $r_{1},\ldots ,r_{l}\in Z_{p}$. The output ciphertext will be as follows:
$$\begin{array}{ccc} CT & = & \{M\cdot e{\left(g,g\right)}^{\alpha s},g^{s},\left(g^{a\lambda_{1}}H{\left(\rho \left(1\right)\right)}^{-r_{1}},g^{r_{1}}\right),\\ & & \ldots ,(g^{a\lambda_{l}}H{\left(\rho \left(l\right)\right)}^{-r_{l}},g^{r_{l}})\}\\ & = & \{C,C^{\prime },\left(C_{1},D_{1}\right),\ldots ,\left(C_{l},D_{l}\right)\}, \end{array}$$
with a description of (M,ρ).KeyGen(MSK,S)→SK: Given set *S* of attributes, this algorithm chooses $t\in Z_{p}$ randomly and outputs the private key as:
$$K=g^{\alpha +at},L=g^{t},\forall x\in SK_{x}=H{\left(x\right)}^{t}.$$Decrypt(CT,SK)→M: Suppose that *S* satisfies the access structure and let I⊂{1,…,l} be defined as I={i:ρ(i)∈S}. This algorithm then finds a set of constants $\left\{w_{i}\in Z_{p}\right\}$ such that $\sum_{i\in I}w_{i}\lambda_{i}=s$. The decryption algorithm computes
$$e\left(C^{\prime },K\right)/\left(\prod_{i\in I}{\left(e\left(C_{i},L\right)e\left(D_{i},K_{\rho (i)}\right)\right)}^{w_{i}}\right)=e{\left(g,g\right)}^{\alpha s}$$
and derives *M* from the ciphertext.

The security of Waters CP-ABE scheme is based on the decisional *q*-parallel bilinear Diffie–Hellman exponent (BDHE) assumption, which is defined as follows:

**Definition** **2**(Decisional *q*-parallel BDHE Assumption)**.**
*Let $a,s,b_{1},\ldots ,b_{q}\overset{R}{\leftarrow }Z_{p}$ and g be a generator of G. Given*
$$D:=\left\{\begin{array}{c} g,g^{s},g^{a},\ldots ,g^{\left(a^{q}\right)},g^{\left(a^{q+2}\right)},\ldots ,g^{\left(a^{2q}\right)}\\ \forall_{1\leq j\leq q}\;\begin{array}{c} g^{s\cdot b_{j}},g^{a/b_{j}},\ldots ,g^{\left(a^{q}/b_{j}\right)},\\ g^{\left(a^{q+2}/b_{j}\right)},\ldots ,g^{\left(a^{2q}/b_{j}\right)} \end{array}\\ \forall_{1\leq j,k\leq q,k\neq j}\;g^{a\cdot s\cdot b_{k}/b_{j}},\ldots ,g^{a^{q}\cdot s\cdot b_{k}/b_{j}} \end{array}\right\}$$
*and element $T\in G_{T}$, we assume that for any probabilistic polynomial time (PPT) algorithm A that outputs in {0,1},*
$$Adv_{A}:=\left|P\left[A\left(D,e{\left(g,g\right)}^{a^{q+1}s}\right)=1\right]-P\left[A\left(D,T\right)=1\right]\right|$$
*is negligible.*

**Theorem** **1.**
*Suppose the decisional q-parallel BDHE assumption holds, then no polynomial-time adversary can selectively break the Waters CP-ABE system in the CPA-model.*


In this paper, we also use this access structure to build our ABE system. Waters CP-ABE scheme is semantically secure when the decisional *q*-parallel BDHE assumption holds. More construction details and the security proof can be found in Waters’ work [14].

#### 2.3.3. Composite Order Bilinear Groups

The composite order bilinear group was first introduced by Boneh et al. [33]. Let G and $G_{T}$ be two multiplicative cyclic groups of composite order $N=p_{1}p_{2}\ldots p_{m}$, where $p_{1},p_{2},\ldots ,p_{m}$ are distinct primes, with a bilinear map function $e:G\times G\to G_{T}$. G has a subgroup $G_{p_{i}}$ of order $p_{i}$ for each prime $p_{i}$. Let $g_{1},g_{2},\ldots ,g_{m}$ be the generators of these subgroups respectively. Each element in G can be expressed in the form $g_{1}^{a_{1}}g_{2}^{a_{2}}\ldots g_{m}^{a_{m}}$, where $a_{1},a_{2},\ldots ,a_{m}\in Z_{N}$.

Orthogonality is an important property of the composite bilinear groups. If $u\in G_{p_{i}},v\in G_{p_{j}}$ and i≠j, then e(u,v)=1, where 1 is the identity element in $G_{T}$.

There is a complexity assumption in the composite group called the subgroup decision assumption. In ref. [34], the definition is given as follows:

**Definition** **3**(General Subgroup Decision Assumption)**.**
*Let $S_{0},S_{1},S_{2},\ldots ,S_{k}$ be non-empty subsets of 1,…,m such that for each 2≤j≤k, either $S_{j}\cap S_{0}=\emptyset =S_{j}\cap S_{1}$ or $S_{j}\cap S_{0}\neq \emptyset \neq S_{j}\cap S_{1}$. Given group generator G, we define the following distribution:*
$$\begin{array}{c} PP:=\left\{N=p_{1}p_{2}\ldots p_{m},G,G_{T},e\right\}\overset{\;R\;}{\leftarrow }G\\ Z_{i}\overset{\;R\;}{\leftarrow }G_{S_{i}}\forall i\in \left\{1,\ldots ,k\right\},\\ D:=\{PP,Z_{2},\ldots ,Z_{k}\}. \end{array}$$
*We assume that for any PPT algorithm A with output in {0,1},*
$$Adv_{G,A}:=\left|P\left[A\left(D,Z_{0}\right)=1\right]-P\left[A\left(D,Z_{1}\right)=1\right]\right|$$
*is negligible.*


This assumption implies that it is hard to determine whether two given elements are composed of the same subgroups when they contain at least one common subgroup. In this work, we utilize this property to construct a deniable ciphertext that is indistinguishable to a normal ciphertext.

#### 2.3.4. Chameleon Hash

The chameleon hash scheme, which was first introduced by Krawczyk et al. [35], uses trapdoor pseudo-random permutation functions as one-way functions. Without the trapdoor, a pseudo-random permutation function can output random results that satisfy the collision resistance and semantic security of a one-way hash function. With the trapdoor, it is easy to generate collisions. Most public-key encryption systems can be used as trapdoor pseudo-random permutation functions. That is, it is easy to construct a chameleon hash function from a public key system. The input of a chameleon hash has two parts, the input message *m* and the random string *r*. The random string *r* is a parameter used to provide an opportunity to generate a collision. There are three algorithms in a chameleon hash scheme, as defined below.
Setup$\left(1^{\lambda }\right)\to \left\{PK,SK\right\}$: Given a security parameter, the scheme outputs public parameter PK and secret trapdoor SK.Hash(PK,m,r)→h: An efficient and probabilistic algorithm, with inputs PK, a message *m*, and a random string *r*, outputs a hash value *h*.Forgery$\left(PK,SK,m_{0},r_{0},m_{1}\right)\to r_{1}$: An efficient and probabilistic algorithm, with a given message $m_{0}$, a random $r_{0}$, the trapdoor SK and another message $m_{1}$, outputs a random string $r_{1}$ that satisfies the following equation:
$$CH\left(m_{0},r_{0}\right)=CH\left(m_{1},r_{1}\right).$$

There are three associated requirements, collision resistance, semantic security and collision forgery, are listed below.

**Definition** **4**(Collision Resistance)**.**
*Given a chameleon hash scheme {PK,SK,CH(·,·)}, where PK is the public information, SK is the trapdoor and CH(·,·) is the hash function. Let $m,m^{\prime }$ be two different messages and r a random string. We call the scheme collision resistant if for any PPT algorithm A, it is hard to output $r^{\prime }$ such that $CH\left(m,r\right)=CH\left(m^{\prime },r^{\prime }\right)$ without SK.*

**Definition** **5**(Semantic Security)**.**
*Given a chameleon hash scheme {PK,SK,CH(·,·)}, where PK is the public information, SK is the trapdoor and CH(·,·) is the hash function. We call the scheme semantically secure if for all pairs of message $m,m^{\prime }$ and random string r, the probability distribution of CH(m,r) and $CH(m^{\prime },r)$ are computationally indistinguishable.*

**Definition** **6**(Collision Forgery)**.**
*Given a chameleon hash scheme {PK,SK,CH(·,·)}, where PK is the public information, SK is the trapdoor and CH(·,·) is the hash function. Let $m,m^{\prime }$ be two different messages and r is a random string. We call the scheme a collision forgery scheme if there exists one PPT algorithm A that on input SK, outputs a string $r^{\prime }$ that satisfies $CH\left(m,r\right)=CH\left(m^{\prime },r^{\prime }\right)$.*

In the remainder of this paper, for simplicity, we use CH to denote the chameleon hash public information and CH(·,·) to denote the chameleon hash operation.

## 3. Privacy-Preserving Broker-ABE Scheme

### 3.1. Overview and Attack Model

The privacy-preserving broker-ABE scheme includes four roles, the trusted key server, the sender, the broker, and the receiver. The key server generates the encryption environment and related keys. The sender, which is usually a sensor in the MCCPS scenario, is in charge of data encryption and forwarding ciphertexts to a broker. The broker re-encrypts the received ciphertexts by embedding policies according to the sender’s identity. Then the broker sends the ciphertexts to all possible receivers, which are CSPs in the MCCPS. If a CSP’s attributes satisfy the policy setup by the broker, the CSP can successfully decrypt the ciphertexts and process the data collected by senders. This procedure is illustrated in Figure 2.

Now we describe how the attacker can attack this system. In our scheme, we focus on two attack ways. First, we assume that an attacker can intercept all encrypted data, including data between the sensor and the broker and between the broker and the CSP. So the transmission data must be encrypted confidentially. Those who cannot satisfy the ciphertext policy should not be able to decrypt the message. Second, we assume that the attacker can force the CSP to release the user data with a proof to show that the data is decrypted from the ciphertext. Undoubtedly, the most trivial proof is the CSP key so the attacker can decrypt the ciphertext itself. However, the user data will be released to the attacker. We want to protect the CSP under this attack scenario.

The first attack is common for most encryption schemes. The security model used in this paper is described in Section 3.3. As for the second attack way, we apply the concept of deniable encryption. We make the CSP convince the attacker with fake data so the real data is kept secret. The key point is that the fake data must be verified with the ciphertext captured by the attacker.

In this paper, we use the multi-distributional deniable encryption approach. The idea of the multi-distributional deniable encryption scheme is to use two different sets of algorithms. One is the normal set and the other is the deniable set. The outputs generated from these two sets are computationally indistinguishable. So the whole system claims to use the normal set while the system actually uses the deniable set and the attacker cannot challenge the claim. In the deniable set, the sender can embed a fake message in the encryption step. Since the ciphertext is indistinguishable to the ciphertext generated from the normal encryption algorithm, we can claim the ciphertext is normally encrypted and there is no fake data. The process can be shown in Figure 3.

We simply use the secret key in the CSP as the proof, so we use the decryption algorithm as the verification function.

Note that in our design, the broker will not be allowed to participate in the deniable process.

### 3.2. Definition

The notations listed in Table 1 are applied throughout this paper.

The algorithms of the privacy-preserving broker-ABE scheme are described below:Setup$\left(1^{\lambda }\right)\to \left(PP,MSK\right)$: This algorithm takes security parameter λ as input and returns public parameter PP and system master key MSK.Setup${}_{B}\left(PP,MSK,ID_{B}\right)\to \left(PP_{B},MSK_{B}\right)$: This algorithm takes the broker ID and system-wise information as input and outputs broker-wise public parameter $PP_{B}$ and broker-wise master key $MSK_{B}$. Note that this information is only for the broker and the sensors under that broker. This algorithm enables the whole system to simultaneously support multiple brokers.KeyGen${}_{1}$$\left(MSK_{B}\right)\to SK_{1}$: This algorithm takes $MSK_{B}$ as input and returns $SK_{1}$ for the broker.KeyGen${}_{2}$$\left(MSK,S\right)\to SK_{2}$: Given an attribute set *S* of a CSP and MSK, this algorithm outputs private key SK for the CSP.Enc${}_{1}$$(PP_{B},M)\to C$: The phase 1 encryption is performed in the sender. Given the message *M* and $PP_{B}$, the sender can output a ciphertext *C* that will be delivered to the broker.Enc${}_{2}$$\left(PP,C,SK_{1},A\right)\to C^{*}$: The phase 2 encryption is performed in the broker. Given the ciphertext *C*, PP and LSSS access structure *A*, the broker can re-encrypt *C* to a new ciphertext $C^{*}$.Dec$\left(PP,SK_{2},C^{*}\right)\to \left\{M,⊥\right\}$: The decryption is performed in the CSP. If *S* satisfies *A*, the decryption algorithm returns *M*; otherwise, ⊥.OpenDec$\left(PP,SK_{2},C^{*},M\right)\to P_{D}$: The algorithm is used to release evidence for proving that $C^{*}$ is the encryption result from *M*.DenSetup$\left(1^{\lambda }\right)\to \left(PP,MSK,PK\right)$: This is the deniable version of Setup algorithm. Except for PP and MSK, the algorithm also generates the system public information PK. PK is known by all sensors and CSPs, and is kept secret from outsiders. Note that the broker is not in charge of the deniable part so PK is also kept secret from the broker.DenKeyGen${}_{2}\left(MSK,PK,S\right)\to \left(SK_{2},SK_{2}^{\prime }\right)$: This is the deniable version of KeyGen algorithm. Except for $SK_{2}$, which is derived from the normal function, the algorithm also returns a fake key $SK_{2}^{\prime }$, which is used later to generate fake proof.DenEnc${}_{1}\left(PP_{B},PK,M,M^{\prime },A\right)\to C^{\prime }$: Except for the inputs of the normal encryption algorithm, this deniable encryption algorithm requires public key PK and a pre-defined fake message $M^{\prime }$. The output ciphertext $C^{\prime }$ must be indistinguishable from the output of Enc.DenOpenDec$\left(PP,SK,SK_{2}^{\prime },C^{*},M^{\prime }\right)\to P_{D}^{\prime }$: Compared to the normal version algorithm, the algorithm takes additional inputs FK and $M^{\prime }$. The output is evidence that $C^{*}$ is from $M^{\prime }$ instead of *M*.

Note that the broker cannot see the message in plaintext form so will not participate in the open process. Therefore, there is no DenEnc${}_{2}$ and DenKeyGen${}_{1}$. Because of the storage constraint issue, the sensor stores no historical data and the deniability is only designed for the CSP. For this reason, there are no OpenEnc and DenOpenEnc algorithms.

For each algorithm, we use Table 2 to point out which entity should run it.

The following properties are required in the proposed scheme:Security: the outputs of Enc${}_{1}$ and Enc${}_{2}$ must be proved to be confidential under the security model, which is described in Section 3.3. The first protects the segment between the sensor and the broker, whereas the latter protects the segment between the broker and the CSP. The security of Enc${}_{1}$ also maintains secrecy from the broker. Note that here we do not mention the security of DenEnc${}_{1}$ because the outputs of Enc${}_{1}$ and DenEnc${}_{1}$ must be indistinguishable. If one is secure and the other is not, this will make these two algorithms distinguishable. So, here, the indistinguishability proof also implies the security proof.Deniability: the proposed scheme is a receiver-DE scheme. That is, given public parameter PP, the two distribution tuples $\left(M,C,P_{D}\right)$ and $\left(M^{\prime },C^{\prime },P_{D}^{\prime }\right)$ are computationally indistinguishable, where $M,M^{\prime }$ are claimed messages, $C,C^{\prime }$ are normally and deniably encrypted ciphertexts, respectively, and $P_{D},P_{D}^{\prime }$ are proofs generated from the normal and deniable open algorithms, respectively. That is, there is no PPT algorithm *A* for which
$$Adv_{A}:=\left|\begin{array}{cc} & P[A\left(PP,\left(M,C,P_{D}\right)\right)=1]\\ - & P[A\left(PP,\left(M^{\prime },C^{\prime },P_{D}^{\prime }\right)\right)=1] \end{array}\right|$$
is non-negligible. Note that the deniably encrypted ciphertexts are the outputs of Enc${}_{2}$ since no entity should be able to decrypt the outputs of Enc${}_{1}$ or DenEnc${}_{1}$. Deniability in the sensor is not considered since it makes no sense for a sensor to provide lots of storage for evidence.Deniable proof consistency: in the MCCPS scenario, a cloud service may be used by many sensors. Some sensors may use the privacy-preserving encryption scheme and some may not. When releasing a deniable proof of a CSP, the proof should look convincing not only to the sensor that uses the DE algorithms, but also to the sensor that uses the normal encryption algorithms. That is, given a set of ciphertexts C, including normally encrypted ciphertexts and deniably encrypted ciphertexts, normal proof $P_{D}$ and deniable proof $P_{D}^{\prime }$, there is no PPT algorithm *A* for which
$$Adv_{A}:=\left|P\left[A\left(C,P_{D}\right)=1\right]-P\left[A\left(C,P_{D}^{\prime }\right)=1\right]\right|$$
is non-negligible.

### 3.3. Security Model

In this subsection, we define the security model used in this scheme. First, an adversary is given a challenge question and is allowed to query an oracle for some information. The adversary wins the game if it can correctly answer the question. The formal security game is as follows:Setup: the challenger first runs Setup and outputs PP to the adversary.Phase 1: the adversary generates queries $q_{1},\ldots ,q_{m}$ for the challenger. Query $q_{i}$ can be one of the following two types of queries:
–Key query: the adversary asks the challenger a key for some entity and obtains its private key from the challenger.–Decryption query: the adversary asks the challenger to decrypt ciphertext $C_{i}$ and obtains its plaintext.Challenge: the adversary chooses two plaintexts $M_{0},M_{1}$ for the challenger. The adversary also provides a challenge condition $A^{*}$, which cannot be authorized by the entities used in $q_{1},\ldots ,q_{m}$. The challenger randomly chooses one bit b∈{0,1} and encrypts the message via Enc$\left(PP,A^{*},M_{b}\right)\to C^{*}$. The challenger sends $C^{*}$ to the adversary as the challenge ciphertext.Phase 2: As in Phase 1, the adversary generates queries $q_{m+1},\ldots ,q_{n}$ for the challenger. Query $q_{i}$ can be one of the following two types of queries:
–Key query: the adversary asks the challenger a key for some entity and obtains its private key from the challenger. Note that the entity cannot be the one who is authorized to decrypt the challenge ciphertext.–Decryption query: the adversary asks the challenger to decrypt ciphertext $C_{i}$ and obtains its plaintext. $C_{i}$ cannot be $C^{*}$.Guess: The adversary returns guess result $b^{\prime }\in \left\{0,1\right\}$. The adversary wins if $b^{\prime }=b$.

The advantage is defined as $|P\left(b^{\prime }=b\right)-\frac{1}{2}|$.

Note that the proposed scheme include two encryption algorithms Enc${}_{1}$ and Enc${}_{2}$. The above definition can be applied to both encryption algorithms (though that there is actually no key can be used to decrypt the output of Enc${}_{1}$. So we will prove Enc${}_{1}$ to be semantic secure in Section 4. Note that semantic security is equal to CPA-security). The entity in Enc${}_{2}$ can be treated as a set of attributes and the condition can be treated as the policy.

**Definition** **7.**
*An encryption scheme is CPA secure if all polynomial-time adversaries have at most a negligible advantage in the above game without any decryption queries.*


**Definition** **8.**
*An encryption scheme is CCA secure if all polynomial-time adversaries have at most a negligible advantage in the above game.*


Note that in the privacy-preserving broker-ABE scheme, there are two encryption algorithms, Enc${}_{1}$ and Enc${}_{2}$. Both output ciphertexts must be verified in the security model.

### 3.4. Assumptions

In this subsection, we describe some non-computational assumptions in this scheme. First, there is no traitor on either the sensor side or the CSP side. DE is a technique in which both the sender and receiver agree to cheat outsiders to protect privacy. If anyone refuses to apply this technique and releases real data to any third party, the whole system will collapse. That is, in this paper, we do not consider the issues of a compromising sensor or the CSP leakage.

Second, according to the definition above, the CSP and the sensor share one semi-public secret PK. This information is shared by every entity that uses the privacy preservation service but is confidential to outsiders. Here, we assume the existence of an authentication mechanism to determine whether a new applicant is a spy. If PK is leaked to an outsider, the outsider is then enabled to answer whether a released proof is normal or deniable by generating a deniable ciphertext. So, we must require PK to be unknown to adversaries.

Third, we assume that the broker will honestly re-encrypt ciphertexts. In this work, we assume the broker is semi-trusted which means the broker cannot see the data content. However, the broker is responsible for embedding the policy checking mechanism into the ciphertexts according to the sensor types. Although the broker cannot decrypt the ciphertexts, it may embed the wrong policy and launch other attacks, e.g., denial-of-service. In this work, we assume the broker is honest-but-curious.

### 3.5. Construction

The proposed scheme, which is based on Waters’s scheme [14], is as follows.
Setup$\left(1^{\lambda }\right)\to \left(PP,MSK\right)$: this algorithm generates a bilinear group G of order $N=p_{1}p_{2}p_{3}$, where $p_{1},p_{2},p_{3}$ are distinct primes with the bilinear map function $e:G\times G\to G_{T}$. Note that $G_{T}$ is also order *N*. G can be separated into three orthogonal subgroups $G_{p_{1}},G_{p_{2}},G_{p_{3}}$. This algorithm picks three generators $g_{1},g_{2},g_{3}$ for $G_{p_{1}},G_{p_{2}},G_{p_{3}}$ respectively. The algorithm also randomly picks $a,\beta \in Z_{N}$ and $\alpha_{1}\in Z_{p_{1}}$ and chooses a hash function $H_{1}:{\left\{0,1\right\}}^{*}\to G_{p_{1}}$. The output will be:
$$\begin{array}{c} PP=\{G,e,H_{1},g_{1},g_{2},g_{1}^{a},e{\left(g_{1},g_{1}\right)}^{\alpha_{1}},e{\left(g_{1},g_{1}\right)}^{\beta }\},\\ MSK=\{\alpha_{1},\beta \}. \end{array}$$Setup${}_{B}\left(PP,MSK,ID_{B}\right)\to \left(PP_{B},MSK_{B}\right)$: taking PP and MSK as inputs, the algorithm generates a new public information and a new master key in a broker domain. $ID_{B}$ is the broker identity and acts an index for recording the secret of a broker. The algorithm first randomly picks $\alpha_{2}\in Z_{p_{2}}$. The output will be:
$$\begin{array}{cc} PP_{B}= & \{G,e,H_{1},g_{1},g_{2},g_{1}^{a},e{\left(g_{1},g_{1}\right)}^{\beta },\\ & e{\left(g_{1},g_{1}\right)}^{\alpha_{1}},e{\left(g_{2},g_{2}\right)}^{\alpha_{2}}\},\\ MSK_{B}= & \{\alpha_{1},\alpha_{2},\beta \}. \end{array}$$KeyGen${}_{1}$$\left(MSK_{B}\right)\to SK_{1}$: the algorithm generates $SK_{1}$ for the broker as follows:
$$SK_{1}=\left\{g_{2}^{\alpha_{2}}\right\}.$$KeyGen${}_{2}$$\left(MSK,S\right)\to SK_{2}$: the algorithm generates $SK_{2}$ for the CSP based on its attributes *S*. It chooses $t\in Z_{N}$ randomly and outputs private key $SK_{2}$ as follows:
$$\begin{array}{cc} SK_{2} & =\{g_{1}^{\alpha_{1}},g_{1}^{\beta +at},g_{1}^{t},{\left\{H_{1}{\left(x\right)}^{t}\right\}}_{\forall x\in S}\}\\ & =\{K,K^{*},L,{\left\{K_{x}\right\}}_{\forall x\in S}\}. \end{array}$$
Note that, in this system, each CSP will share the same *K*.Enc${}_{1}$$(PP_{B},M)\to C$: given message *M*, the algorithm randomly picks $s\in Z_{N}$. Then, the algorithm sets up a one-way hash function *H*. Note that hash function *H* can be any kind of one-way function, including pseudo random permutation functions. Next, the algorithm flips two coins $b_{0},b_{1}$ and selects two random strings $t_{0},t_{1}$. The output ciphertext *C* will be:
$$C=\{A_{0},A_{1},B,H,t_{0},t_{1},V\},$$
where,
$$\begin{array}{cc} A_{b_{0}} & =M\cdot e{\left(g_{1},g_{1}\right)}^{\alpha_{1}s}e{\left(g_{2},g_{2}\right)}^{\alpha_{2}s},\\ A_{1-b_{0}} & \overset{R}{\leftarrow }G_{T},\\ B & ={\left(g_{1}g_{2}\right)}^{s},\\ V & =H(M,t_{b_{1}})\\ & \neq H(A_{1-b_{0}}\cdot e{\left(g_{1},g_{1}\right)}^{-\alpha_{1}s}e{\left(g_{2},g_{2}\right)}^{-\alpha_{2}s},\\ & t_{1-b_{1}}). \end{array}$$Enc${}_{2}$$\left(PP,C,SK_{1},A\right)\to C^{*}$: given a ciphertext *C* and an LSSS access structure A, the algorithm first removes the $G_{p_{2}}$ part in $A_{0}$ and $A_{1}$ as follows:
$$A_{b}^{\prime }=A_{b}\cdot e{\left(B,g_{2}^{\alpha_{2}}\right)}^{-1}=M\cdot e{\left(g_{1},g_{1}\right)}^{\alpha_{1}s},\forall b=\left\{0,1\right\}.$$
Let A=(M,ρ) where M is a l×n matrix and ρ is a mapping function from {1,…,l} to the attribute field. The algorithm randomly generates two vectors $\vec{v}=\left(s^{*},y_{2},\ldots ,y_{n}\right)\in Z_{N}^{n}$ and $\vec{r}=\left(r_{1},\ldots ,r_{l}\right)\in Z_{N}^{l}$. Then it calculates $\lambda_{i}=\vec{v}M_{i},\forall i\in \left\{1,\ldots ,l\right\}$. The output ciphertext $C^{\prime }$ will be as follows:
$$\begin{array}{cc} C^{*}=\{ & A_{0}^{*},A_{1}^{*},B,B^{*},\left(C_{1},D_{1}\right),\ldots ,\left(C_{l},D_{l}\right),\\ & H,t_{0},t_{1},V\}, \end{array}$$
where
$$\begin{array}{c} A_{b}^{*}=A_{b}^{\prime }\cdot e{\left(g_{1},g_{1}\right)}^{\beta s^{*}},\forall b\in \left\{0,1\right\},\\ B^{*}=g_{1}^{s^{*}},\\ C_{i}=g_{1}^{a\lambda_{i}}H_{1}{\left(\rho \left(i\right)\right)}^{-r_{i}},D_{i}=g_{1}^{r_{i}},i=1\ldots l, \end{array}$$
$B,H,t_{0},t_{1},V$ are directly derived from *C* and are not changed. A is also appended to $C^{*}$.Dec$\left(PP,SK_{2},C^{*}\right)\to \left\{M,⊥\right\}$: to decrypt ciphertext $C^{*}$ for access structure A, the algorithm first checks if attribute set *S* of $SK_{2}$ satisfies A. Suppose *S* satisfies A and let I⊂{1,2,…,l} be defined as I={i:ρ(i)∈S}. The algorithm is able to find a set of constants $\left\{w\in Z_{N}\right\}$ such that $\sum_{i\in I}w_{i}\lambda_{i}=s^{*}$. This algorithm computes $M_{0},M_{1}$ as follows:
$$M_{b}=A_{b}^{*}\cdot \frac{\prod_{i\in I}{\left(e\left(C_{i},L\right)e\left(D_{i},K_{\rho (i)}\right)\right)}^{w_{i}}}{e\left(B,K\right)e\left(B^{*},K^{*}\right)},\forall b\in \left\{0,1\right\}.$$
The algorithm then checks tag *V* as follows:
$$v_{i,j}=H\left(M_{i},t_{j}\right),\forall i,j\in \left\{0,1\right\}.$$
If $v_{i,j}$ is equal to *V*, then $M_{i}$ is a true message. Otherwise, this algorithm returns ⊥.OpenDec$\left(PP,SK_{2},C^{*},M\right)\to SK_{2}$: to show that $C^{*}$ is encrypted from *M*, the algorithm simply returns $SK_{2}$.DenSetup$\left(1^{\lambda }\right)\to \left(PP,MSK,PK\right)$: (PP,MSK) tuple is generated as Setup. The algorithm also randomly picks $\alpha_{3}\in Z_{p_{3}}$. PK is generated as follows:
$$PK=\{g_{3},e{\left(g_{3},g_{3}\right)}^{\alpha_{3}}\}.$$
Note that PK is kept secret from outsiders, including the broker, since it does not participate in the deniable encryption process. For the same reason, there is no DenSetup${}_{B}$ algorithm.DenKeyGen${}_{2}$$\left(MSK,PK,S\right)\to \left(SK_{2},SK_{2}^{\prime }\right)$: $SK_{2}$ is derived from KeyGen and $SK_{2}^{\prime }$ is generated as follows:
$$\begin{array}{cc} SK_{2}^{\prime } & =\{g_{1}^{\alpha_{1}}g_{3}^{\alpha_{3}},g_{1}^{\beta +at},g_{1}^{t},{\left\{H_{1}{\left(x\right)}^{t}\right\}}_{\forall x\in S}\}\\ & =\{K^{\prime },K^{*},L,{\left\{K_{x}\right\}}_{\forall x\in S}\}. \end{array}$$DenEnc${}_{1}$$(PP,PK,M,M^{\prime })\to C$: the algorithm needs one more input $M^{\prime }$ which is the pre-determined fake message (that is, a fake message is given by users. If the fake message is obviously far from the normal message, it is the user’s responsibility rather than that of the proposed scheme). This algorithm first runs Enc and gets $\lambda_{i},\forall i\in \left\{1,\ldots ,l\right\}$ and $b_{0},b_{1}$. Next, this algorithm sets up a chameleon hash function CH(·,·). The output deniable ciphertext *C* will be:
$$C^{\prime }=\left\{A_{0},A_{1},B,CH,t_{0},t_{1},V\right\},$$
where,
$$\begin{array}{c} A_{b_{0}}=M\cdot e{\left(g_{1},g_{1}\right)}^{\alpha_{1}s}e{\left(g_{2},g_{2}\right)}^{\alpha_{2}s},\\ A_{1-b_{0}}=M^{\prime }\cdot e{\left(g_{1},g_{1}\right)}^{\alpha_{1}s}e{\left(g_{2},g_{2}\right)}^{\alpha_{2}s}e{\left(g_{3},g_{3}\right)}^{\alpha_{3}s},\\ B={\left(g_{1}g_{2}g_{3}\right)}^{s},\\ V=CH\left(M,t_{b_{1}}\right)=CH\left(M^{\prime },t_{1-b_{1}}\right). \end{array}$$
Since a chameleon hash function is a trapdoor pseudo-random permutation function, it is easy to find $t_{b_{1}},t_{1-b_{1}}$ to generate a collision, which implies that *M* and $M^{\prime }$ can both be valid in decryption.DenOpenDec$\left(PP,SK_{2}^{\prime },FK,C^{*},M^{\prime }\right)\to SK_{2}^{\prime }$: to show that $C^{*}$ is encrypted from the fake message $M^{\prime }$, the algorithm simply returns $SK_{2}^{\prime }$.

In this construction, the sensor is in charge of Enc${}_{1}$ and the broker is in charge of Enc${}_{2}$ for embedding policy re-encryption. There is no deniable encryption feature for Enc${}_{2}$ since the broker knows nothing about the message, which makes it impossible for the broker to create fake messages. There is only one decryption algorithm because outsiders may run Dec with keys derived from CSPs to verify message contents. Note that with the correct key, no matter which encryption algorithm is used, the CSP can always get the correct message. With respect to the fake key, the outsider will get fake messages if they are deniably encrypted. However, if a message is normally encrypted, the real message can also be derived via the fake key.

### 3.6. Correctness

In this subsection, we check the correctness of this construction. Since only the receiver can correctly decrypt the ciphertext, the correction check is only invoked for ciphertext *C* and $SK_{2}$. Below, we present four scenarios and their corresponding checks.
When using the normal key $SK_{2}$ to decrypt a normally encrypted ciphertext *C*, the decryption process will be as follows:
$$\begin{array}{cc} & \frac{\prod_{i\in I}{\left(e\left(C_{i},L\right)e\left(D_{i},K_{\rho (i)}\right)\right)}^{w_{i}}}{e\left(B,K\right)e\left(B^{*},K^{*}\right)}\\ = & \frac{\prod_{i\in I}{\left(e\left(g_{1}^{a\lambda_{i}},g_{1}^{t}\right)\right)}^{w_{i}}}{e\left({\left(g_{1}g_{2}\right)}^{s},g_{1}^{\alpha_{1}}\right)e\left(g_{1}^{s^{*}},g_{1}^{\beta +at}\right)}\\ = & \frac{e{\left(g_{1},g_{1}\right)}^{at\prod_{i\in I}\lambda_{i}w_{i}}}{e{\left(g_{1},g_{1}\right)}^{s\alpha_{1}+s^{*}\left(\beta +at\right)}}\\ = & e{\left(g_{1},g_{1}\right)}^{-\alpha_{1}s-\beta s^{*}}. \end{array}$$
With the hash function *H* and *V*, the receiver can correctly get the message *M*.When using the normal key $SK_{2}$ to decrypt a deniably encrypted ciphertext *C*, the decryption process will be as follows:
$$\begin{array}{cc} & \frac{\prod_{i\in I}{\left(e\left(C_{i},L\right)e\left(D_{i},K_{\rho (i)}\right)\right)}^{w_{i}}}{e\left(B,K\right)e\left(B^{*},K^{*}\right)}\\ = & \frac{\prod_{i\in I}{\left(e\left(g_{1}^{a\lambda_{i}},g_{1}^{t}\right)\right)}^{w_{i}}}{e\left({\left(g_{1}g_{2}g_{3}\right)}^{s},g_{1}^{\alpha_{1}}\right)e\left(g_{1}^{s^{*}},g_{1}^{\beta +at}\right)}\\ = & \frac{e{\left(g_{1},g_{1}\right)}^{at\prod_{i\in I}\lambda_{i}w_{i}}}{e{\left(g_{1},g_{1}\right)}^{s\alpha_{1}+s^{*}\left(\beta +at\right)}}\\ = & e{\left(g_{1},g_{1}\right)}^{-\alpha_{1}s-\beta s^{*}}. \end{array}$$
With the chameleon hash function CH and *V*, the receiver can correctly get the message *M*.When using the deniable key $SK_{2}^{\prime }$ to decrypt a deniably encrypted ciphertext *C*, the decryption process will be as follows:
$$\begin{array}{cc} & \frac{\prod_{i\in I}{\left(e\left(C_{i},L\right)e\left(D_{i},K_{\rho (i)}\right)\right)}^{w_{i}}}{e\left(B,K^{\prime }\right)e\left(B^{*},K^{*}\right)}\\ = & \frac{\prod_{i\in I}{\left(e\left(g_{1}^{a\lambda_{i}},g_{1}^{t}\right)\right)}^{w_{i}}}{e\left({\left(g_{1}g_{2}g_{3}\right)}^{s},g_{1}^{\alpha_{1}}g_{3}^{\alpha_{3}}\right)e\left(g_{1}^{s^{*}},g_{1}^{\beta +at}\right)}\\ = & \frac{e{\left(g_{1},g_{1}\right)}^{at\prod_{i\in I}\lambda_{i}w_{i}}}{e{\left(g_{1},g_{1}\right)}^{s\alpha_{1}+s^{*}\left(\beta +at\right)}e{\left(g_{3},g_{3}\right)}^{\alpha_{3}s}}\\ = & e{\left(g_{1},g_{1}\right)}^{-\alpha_{1}s-\beta s^{*}}e{\left(g_{3},g_{3}\right)}^{-\alpha_{3}s}. \end{array}$$
With the chameleon hash function CH and *V*, the receiver can correctly get message $M^{\prime }$ rather than the real message *M*.When using the deniable key $SK_{2}^{\prime }$ to decrypt a normally encrypted ciphertext *C*, the decryption process will be as follows:
$$\begin{array}{cc} & \frac{\prod_{i\in I}{\left(e\left(C_{i},L\right)e\left(D_{i},K_{\rho (i)}\right)\right)}^{w_{i}}}{e\left(B,K^{\prime }\right)e\left(B^{*},K^{*}\right)}\\ = & \frac{\prod_{i\in I}{\left(e\left(g_{1}^{a\lambda_{i}},g_{1}^{t}\right)\right)}^{w_{i}}}{e\left({\left(g_{1}g_{2}\right)}^{s},g_{1}^{\alpha_{1}}g_{3}^{\alpha_{3}}\right)e\left(g_{1}^{s^{*}},g_{1}^{\beta +at}\right)}\\ = & \frac{e{\left(g_{1},g_{1}\right)}^{at\prod_{i\in I}\lambda_{i}w_{i}}}{e{\left(g_{1},g_{1}\right)}^{s\alpha_{1}+s^{*}\left(\beta +at\right)}}\\ = & e{\left(g_{1},g_{1}\right)}^{-\alpha_{1}s-\beta s^{*}}. \end{array}$$
With the hash function *H* and *V*, the receiver can correctly get the message *M*.

Based on the above checking process, this scheme has two important properties. First, if a CSP uses the normal key, whether the sensor normally or deniably encrypts the data, the CSP can always derive the correct message. Second, with the deniable key, the CSP can get the correct message even if the sensor normally encrypts the data. That is, if the CSP supports both privacy-preserving and normal sensors, an outsider cannot challenge the key released from the CSP since the key looks normal, even when the sensor does not participate in the deniability process.

**Theorem** **2.**
*The privacy-preserving broker ABE scheme is deniable proof consistent.*


**Proof of Theorem** **2.**In this scheme, the CSP key is treated as proof for the CSP claim. Note that the key is the most immediate proof available. As shown above, both $SK_{2}$ and $SK_{2}^{\prime }$ can be applied to correctly decrypt ciphertexts. That is, a ciphertext can be decrypted to a meaningful message, which may be real data or a pre-determined fake message. Given a set of ciphertexts C where its element may be normally or deniably encrypted, anyone who can differentiate between $\left(C,SK_{2}\right)$ and $\left(C,SK_{2}^{\prime }\right)$ can also differentiate between real and fake data. In other words, these two tuples are indistinguishable.  □

### 3.7. Implementation Issues

In this subsection, we will describe some implementation issues. First, we focus on the performance issue about composite order bilinear groups. Then, we discuss how to run Enc${}_{1}$ and Enc${}_{2}$ on the sensor and the broker efficiently. Note that we do not care about the CSP performance issue because the CSP resource is generally much greater than the sensor and the broker.

In our construction, we use composite order bilinear groups because of their canceling property. However, the composite order bilinear group operations are prolonged. Table 3 shows the comparison result of the bilinear operation on a raspberry PI3, where each prime is 512 bits. It shows that the required operation time grows exponentially. Therefore, some researches suggest that the composite order group is unlikely applicable [36,37].

To solve this problem, in our implementation we used a prime order group to simulate composite order group behavior, as described in Lewko’s work [34]. The idea is based on dual orthonormal bases and the subspace assumption. Each prime subgroup is simulated as a distinct orthonormal base. By the orthogonal property, the bilinear operation will be canceled between different prime subgroups. In our implementation, each base contains three prime order group elements. So the bilinear operation is only around three times than the prime order bilinear operation. Table 3 lists the required times of simulation-based paring operations. We can find that the performance is greatly enhanced.

Some may claim that even with this simulation technique, the bilinear operation is also expensive to a lightweight node like sensors. We emphasize that in our implementation, there is no bilinear operation in the sensor node. According to Table 2, the sensor node only runs Enc and DenEnc. In these two algorithms, $e{\left(g_{1},g_{1}\right)}^{\alpha_{1}}$, $e{\left(g_{2},g_{2}\right)}^{\alpha_{2}}$ and $e{\left(g_{3},g_{3}\right)}^{\alpha_{3}}$ are all public information and can be pre-computed. So the sensor does not need to run any bilinear operations.

Now we focus on the broker part. The broker needs to run one bilinear operation and this operation can be accelerated through the simulation technique. Since the broker is the entity which embeds policies on the ciphertext, some may doubt it takes lots of computational resources and memory, especially with the composite order group. We emphasize that this is not true in our implementation. In this work, we use the composite order group for its canceling property, and we can simulate this property through the prime order group. That is, all elements are in the prime order group. So in Enc${}_{2}$, except for one pairing operation, the only operations are the power operation and the multiplication operation in the prime order group. Moreover, the canceling property is used only between *B*, $SK_{1}$, *K* in $SK_{2}$ and $K^{\prime }$ in $SK_{2}^{\prime }$. So we only need to use orthonormal bases to simulate these elements and the memory cost will not increase too much.

## 4. Evaluation

In this section, we evaluate the proposed scheme in three aspects, security, deniability, and performance.

### 4.1. Security Proof

In this subsection, we will prove the security of our scheme. For simplicity, though we simulate the composite order group with a prime order group in our implementation, here we prove the security with the composite order group. Since this scheme is a re-encryption scheme, it is necessary to show that both Enc${}_{1}$ and Enc${}_{2}$ are secure. Although there is also another encryption algorithm DenEnc${}_{1}$, we need not prove its security because the outputs of Enc${}_{1}$ and DenEnc${}_{1}$ are indistinguishable. The indistinguishability proof is presented in the next subsection. If the output of Enc${}_{1}$ is secure, but the output of DenEnc${}_{1}$ is not, their indistinguishability will be broken. That is, the indistinguishability proof ensures the security of DenEnc${}_{1}$ while Enc${}_{1}$ is secure. First, we show that Enc${}_{1}$ is secure under the discrete logarithm assumption. Then, we prove that Enc${}_{2}$ is secure in the CPA model. We discuss the Enc${}_{2}$ CCA-security issue in Section 5.

**Lemma** **1.**
*Enc${}_{1}$ is semantic secure if the discrete logarithm problem is hard.*


**Proof of Lemma** **1.**To prove the security of Enc${}_{1}$, the focus should be on $e{\left(g_{1},g_{1}\right)}^{\alpha_{1}s}\cdot e{\left(g_{2},g_{2}\right)}^{\alpha_{2}s}$ and ${\left(g_{1}g_{2}\right)}^{s}$. The rest of the parts are verification tags through a one-way function, so they have no information about the message. Therefore, the security proof is the answer to the following question: given $g_{1},g_{2},{\left(g_{1}g_{2}\right)}^{s},e{\left(g_{1},g_{1}\right)}^{\alpha_{1}},e{\left(g_{2},g_{2}\right)}^{\alpha_{2}}$, is finding $e{\left(g_{1},g_{1}\right)}^{\alpha_{1}s}\cdot e{\left(g_{2},g_{2}\right)}^{\alpha_{2}s}$ easy or not? Since *s* can only be derived from ${\left(g_{1}g_{2}\right)}^{s}$, this question is equivalent to a discrete logarithm problem. Although $g_{1}g_{2}$ is a group element that supports the bilinear map operation, there is no $\alpha_{1},\alpha_{2}$ information in $G_{p_{1}p_{2}}$ unless $\alpha_{1},\alpha_{2}$ can be derived from $e{\left(g_{1},g_{1}\right)}^{\alpha_{1}},e{\left(g_{2},g_{2}\right)}^{\alpha_{2}}$, which is also a discrete logarithm problem. So if the discrete logarithm problem is hard, getting $e{\left(g_{1},g_{1}\right)}^{\alpha_{1}s}\cdot e{\left(g_{2},g_{2}\right)}^{\alpha_{2}s}$ is also a hard problem. Therefore, $e{\left(g_{1},g_{1}\right)}^{\alpha_{1}s}\cdot e{\left(g_{2},g_{2}\right)}^{\alpha_{2}s}$ can be treated as a random element in $G_{T}$. So, Enc${}_{1}$ is semantic secure.  □

The above proof demonstrates that the ciphertext encrypted by the sensor is semantic secure. Because semantic security is equal to CPA-security, Enc${}_{1}$ is CPA-secure. That is, the message transferred from the sensor to the broker is secure, even secure from the broker. The next step is to prove Enc${}_{2}$ is CPA-secure, which implies that the communication between the broker and the CSP is secure. When considering Enc${}_{2}$, we skip the Setup${}_{B}$ step and directly integrate Enc${}_{1}$ into Enc${}_{2}$, since these two steps belong to their own broker domain. The attacker now is outside the broker domain and has the system’s public information from Setup. The attacker does not participate in the Enc${}_{1}$ process.

**Lemma** **2.**
*Enc${}_{2}$ is CPA-secure if Waters CP-ABE scheme is CPA-secure.*


**Proof of Lemma** **2.**Let A be an adversary that breaks the above deniable CP-ABE scheme. An algorithm B that can break Waters CP-ABE scheme can be constructed as follows. B is given public parameters through the Waters CP-ABE scheme’s Setup algorithm from challenger X as follows:
$$PP_{W}=\left\{g_{1},g_{1}^{a},e{\left(g_{1},g_{1}\right)}^{\beta }\right\},$$
with prime number $p_{1}$, $G_{p_{1}}$, e(·,·) and $H_{1}\left(\cdot \right)$. For convenience, we use a suffix to represent different subgroups in our proof. Algorithm B proceeds as follows.
Setup: B first picks two different prime numbers $p_{2}$ and $p_{3}$. Next, B generates group G with order $N=p_{1}p_{2}p_{3}$. Note that the subgroup with $p_{1}$ order in G should be the same as $G_{p_{1}}$. B then randomly picks $\alpha_{1}\in Z_{p_{1}}$ and picks a generator $g_{2}$ for $G_{p_{2}}$. B then shows the following to A:
$$PP=\{G,e,H_{1},g_{1},g_{2},g_{1}^{a},e{\left(g_{1},g_{1}\right)}^{\alpha_{1}},e{\left(g_{1},g_{1}\right)}^{\beta }\}.$$
Note that e(·,·) and $H_{1}\left(\cdot \right)$ are the same as the given function from X.Phase 1: when B receives a key generation query for attribute set *S* from A, B simply relays the query to X and obtains $SK_{W}$ as follows:
$$\begin{array}{cc} SK_{W} & =\{g_{1}^{\beta +at},g_{1}^{t},{\left\{H_{1}{\left(x\right)}^{t}\right\}}_{\forall x\in S}\}\\ & =\{K^{*},L,{\left\{K_{x}\right\}}_{\forall x\in S}\}. \end{array}$$
B then generates $g_{1}^{\alpha_{1}}$ and outputs $SK_{2}$ to X as follows:
$$\begin{array}{cc} SK_{2} & =\{g_{1}^{\alpha_{1}},g_{1}^{\beta +at},g_{1}^{t},{\left\{H_{1}{\left(x\right)}^{t}\right\}}_{\forall x\in S}\}\\ & =\{K,K^{*},L,{\left\{K_{x}\right\}}_{\forall x\in S}\}. \end{array}$$Challenge: A outputs two messages $M_{0}$, $M_{1}$ with access structure (M,ρ) to B, and B directly relays $M_{0}$, $M_{1}$ and (M,ρ) to X as the challenge and obtains the following from X.
$$C_{W}=\left\{A^{*},B^{*},\left(C_{1},D_{1}\right),\ldots ,\left(C_{l},D_{l}\right)\right\},$$
where
$$\begin{array}{c} A^{*}=M_{b}\cdot e{\left(g_{1},g_{1}\right)}^{\beta s^{*}},b\in \left\{0,1\right\}\\ B^{*}=g_{1}^{s^{*}},\\ C_{i}=g_{1}^{a\lambda_{i}}H_{1}{\left(\rho \left(i\right)\right)}^{-r_{i}},D_{i}=g_{1}^{r_{i}},i=1\ldots l. \end{array}$$
$M_{b}$ is chosen by X. B setups a chameleon hash function CH and randomly picks $b_{1},b_{2}$ from {0,1}. B also randomly picks $s\in Z_{N}$. Finally, B outputs *C* to A as follows:
$$\begin{array}{cc} C=\{ & A_{0}^{*},A_{1}^{*},B,B^{*},\left(C_{1},D_{1}\right),\ldots ,\left(C_{l},D_{l}\right),\\ & CH,t_{0},t_{1},V\}, \end{array}$$
where,
$$\begin{array}{c} A_{b_{1}}^{*}=A^{*}\cdot e{\left(g_{1},g_{1}\right)}^{\alpha s},\\ A_{1-b_{1}}^{*}\overset{R}{\leftarrow }G_{T},\\ B=g_{1}^{s},\\ V=CH\left(M_{0},t_{b_{2}}\right)=CH\left(M_{1},t_{1-b_{2}}\right). \end{array}$$
Note that a chameleon hash function is used instead of a common hash function. To A, without the trapdoor, a chameleon hash function is simply a one-way function. Here we ensure that the verification tag *V* is valid for both $M_{0}$ and $M_{1}$, so the verification process will lease no information to A.Phase 2: the query and response process is the same as that in Phase 1.Guess: finally, A outputs guess $b^{\prime }$ to B and B forwards guess $b^{\prime }$ to X.If A achieves a non-negligible advantage against the proposed encryption scheme, B can use the output of A to also achieve a non-negligible advantage against the Waters CP-ABE scheme in the CPA model.  □

According to Lemma 1, Lemma 2 and Theorem 1, we can derive the following theorem.

**Theorem** **3.**
*Suppose the discrete logarithm assumption and the decisional q-parallel BDHE assumption hold, then no polynomial time adversary can selectively break the proposed encryption system in the CPA-model.*


### 4.2. Deniability Proof

To prove the deniability of this scheme, we must show that the Enc${}_{1}$ output *C* and the DenEnc${}_{1}$ output $C^{\prime }$, the KeyGen${}_{2}$ output $SK_{2}$ and the DenKeyGen${}_{2}$$SK_{2}^{\prime }$ are indistinguishable respectively. Here we do not consider Enc${}_{2}$ and KeyGen${}_{1}$ because the broker does not participate in the deniability process.

**Lemma** **3.**
*Under the general subgroup decision assumption, normal ciphertext C and deniable ciphertext $C^{\prime }$ are indistinguishable.*


**Proof of Lemma** **3.**Suppose there exists PPT attacker A who achieves a non-negligible advantage in distinguishing the deniable ciphertext $C^{\prime }$ from the normal ciphertext *C* of the proposed scheme. A PPT algorithm B can be constructed that also has a non-negligible advantage against the general subgroup decision assumption.The difference between *C* and $C^{\prime }$ is the existence of the $g_{3}$ element in *B*. Note that there is no difference in $A_{1-b_{0}}$ since in the normal encryption, $A_{1-b_{0}}$ is randomly selected from $G_{T}$ and undoubtedly it is possible to include the $e(g_{3},g_{3})$ element. There is also no difference about the verification parts because CH can be treated as a common hash function if the secret key is not released. Now, given an element *T* to be determined if *T* belongs to $G_{p_{1}p_{2}}$ or $G_{p_{1}p_{2}p_{3}}$. B can construct a ciphertext $C^{*}$ for the two messages *M* and $M^{\prime }$ as follows:
$$C^{*}=\left\{A_{0},A_{1},B,CH,t_{0},t_{1},V\right\},$$
where,
$$\begin{array}{cc} A_{b_{0}} & =M\cdot e{\left(g_{1},T\right)}^{\alpha_{1}}e{\left(g_{2},T\right)}^{\alpha_{2}},\\ A_{1-b_{0}} & =M^{\prime }\cdot e{\left(g_{1},T\right)}^{\alpha_{1}}e{\left(g_{2},T\right)}^{\alpha_{2}}e{\left(g_{3},T\right)}^{\alpha_{3}},\\ B & =T,\\ V & =CH\left(M,t_{b_{1}}\right)=CH\left(M^{\prime },t_{1-b_{1}}\right). \end{array}$$Then B forwards $C^{*}$ to A. If A says it is a normal ciphertext, then *T* is in $G_{p_{1}p_{2}}$. If A says it is a deniable ciphertext, then *T* is in $G_{p_{1}p_{2}p_{3}}$.In the public information PP and $PP_{B}$, there is no $g_{3}$ element. Given the general subgroup decision assumption, it is hard to determine if a given element belongs to $G_{p_{1}p_{2}}$ or $G_{p_{1}p_{2}p_{3}}$. So *C* and $C^{\prime }$ should be indistinguishable.  □

Next, we check the deniability of the released proof.

**Lemma** **4.**
*Under the general subgroup decision assumption, normal decryption proof $SK_{2}$ and deniable decryption proof $SK_{2}^{\prime }$ are indistinguishable.*


**Proof of Lemma** **4.**Suppose there exists PPT attacker A who achieves a non-negligible advantage in distinguishing the deniable proof $SK_{2}^{\prime }$ from the normal proof $SK_{2}$ of the proposed scheme. A PPT algorithm B can be constructed that also has a non-negligible advantage against the general subgroup decision assumption.In this scheme, $SK_{2}$ or $SK_{2}^{\prime }$ used as the receiver proof. The only difference between $SK_{2}$ and $SK_{2}^{\prime }$ is the existence of element $g_{3}$ in *K*. Now given an element *T* to be determined if *T* belongs to $G_{p_{1}}$ or $G_{p_{1}p_{3}}$. B can construct a secret key $SK_{2}^{*}$ as follows:
$$\begin{array}{cc} SK_{2}^{*} & =\{T^{\alpha_{1}},g_{1}^{\beta +at},g_{1}^{t},{\left\{H_{1}{\left(x\right)}^{t}\right\}}_{\forall x\in S}\}\\ & =\{K,K^{*},L,{\left\{K_{x}\right\}}_{\forall x\in S}\}. \end{array}$$Then B forwards $SK_{2}^{*}$ to A. If A says it is a normal key, then *T* is in $G_{p_{1}}$. If A says it is a deniable key, then *T* is in $G_{p_{1}p_{3}}$.In the public information PP and $PP_{B}$, there is no $g_{3}$ element. Given the general subgroup decision assumption, it is hard to determine if a given element belongs to $G_{p_{1}}$ or $G_{p_{1}p_{3}}$. So $SK_{2}$ and $SK_{2}^{\prime }$ should be indistinguishable.  □

From Lemmas 3 and 4, the proposed scheme is a receiver-deniable encryption scheme, which means that outsiders cannot challenge fake keys provided by CSPs.

### 4.3. Performance Evaluation

To evaluate the performance of the proposed scheme, we used a Raspberry Pi 3 computer as the sensor platform and a desktop PC as the broker platform. The PC is equipped with i7-7700 CPU and 16 GB of memory (some may doubt that this PC is too good to be a broker. When considering the fog/edge trend [38], the edge entity is likely to become more and more powerful). In the implementation, we set each prime size to 512 bits, which is equal to 256 bits of security [39]. So, while the composite group order size is 1536 bits, the security level is still 512 bits. We focus on the sensor performance and the broker performance here. The CSP performance is not our concern since the CSP is generally much more powerful.

First, we check the required encryption time in the sensor. We compared our work with Waters scheme [14], which is our base scheme, Yao scheme [23], which is not a pairing scheme, Li scheme [27] (the outsourcing server is a computer with an INTEL i7 CPU. Li et al. separated the encryption work into the data owner and the outsourcing server. Here we combine them together to evaluate the computational cost), which is an outsourcing scheme, and Fischer scheme [28] (Fischer scheme is a KP-ABE scheme), which is a proxy-based scheme like us. We implement our approach in both the composite order group and the prime order group. Figure 4 shows the Enc${}_{1}$ process times. We can find that the attribute number affects the encryption a lot. Only the Fischer scheme and our scheme can keep the same encryption time when the attribute number grows. The main reason is that the attribute related part of the encryption process is outsourced to a powerful entity, and therefore, it will not be counted to the sensor side. So we believe that the broker ABE is suitable for the IoT scenario. Besides, we can find that our prime order simulation implementation improves the encryption process. In our experiment, the exponential operation of G is around 33.66 ms when $N=p_{1}$ and is around 463.95 ms when $N=p_{1}p_{2}p_{3}$. Therefore, we believe that our scheme is affordable for a sensor.

Next, we focused on the transmission cost. We used the ciphertext size to evaluate the transmission cost. Figure 5 is the comparison result. We can find that the ciphertext size has a similar behavior as the encryption time. In most schemes, the ciphertext size grows linear with the attribute number. The ciphertext size of the broker ABE, including our scheme and Fischer scheme, stayed the same since the required attributes are embedded in the broker. Note that the ciphertext size of our scheme was slightly greater than in the Fischer scheme. The reason is that we added a fake element in the ciphertext for data privacy against outside coercion. We believe this is not a problem.

Then we see the Enc${}_{2}$ performance on the broker. Generally speaking, the broker is much more powerful than the sensor, so we only compare our scheme with the Waters scheme. Since we want to evaluate the performance, in our experiments, we make all attributes mandatory so the encryption will be the max encryption time. Again, we implement our scheme in both the composite order and the prime order. Moreover, we implement another version with a trade-off between memory and computation. In our scheme, $\vec{r}$ can be precomputed and therefore $H_{1}{\left(x\right)}^{-r_{i}}$ and $g^{r_{i}}$ can also be precomputed, too. For each x∈S, we make the broker generates γ random numbers and then prepare a pool of $(H_{1}{\left(x\right)}^{-r_{i}},g^{r_{i}}),i\in \left\{1,\ldots ,\gamma \right\}$. So there will be |S| pools. Note that one pair in a pool can be reused because there are total $\gamma^{|S|+1}$ combinations among all pools. We make this trade-off because the broker usually has more memory than the sensor. Figure 6 shows the Enc${}_{2}$ process times. We can see that we greatly decrease the computational loading of the broker.

## 5. Discussion

In this section, we discuss two topics related to this scheme. The first topic is CCA-security, and the second is regarding Chameleon Hash issues.

### 5.1. CCA-Security

In Section 4, we proved that the scheme is CPA-secure. Boneh et al. proved that an IND-sID-CPA secure IBE scheme can be transformed into an IND-sID-CCA secure scheme with the help of a one-time signature scheme [40]. Liang et al. [20] applied this concept and used a random oracle to build a re-encryption scheme that is also based on the Waters scheme. It is trivial to apply this same approach to the scheme proposed in this paper. However, since this is a two-step encryption scheme, there will be two signatures, one by the sensor and the other by the broker. The scheme construction is modified as follows. The algorithms that are not mentioned remain the same as the original construction.
Setup${}_{\mathrm{cca}}\left(1^{\lambda }\right)\to \left(PP,MSK\right)$: Aside from the original Setup, the algorithm additionally setup a hash function $H_{2}:{\left\{0,1\right\}}^{*}\to G_{p_{1}}$ and randomly picks $b\in Z_{N}$. The algorithm then appends $H_{2},g_{1}^{b}$ to PP.Enc${}_{1,\mathrm{cca}}$$\left(PP_{B},M\right)\to C_{\mathrm{cca}}$: The sensor first runs the original Enc${}_{1}$ algorithm and gets $C=\{A_{0},A_{1},B,H,t_{0},t_{1},V\}$. Then the algorithm calculates $B_{1}=g_{1}^{bs}$ and the output ciphertext will be as follows:
$$C_{\mathrm{cca}}=\left\{A_{0},A_{1},B,B_{1},H,t_{0},t_{1},V,V_{1}\right\},$$
where
$$V_{1}=H_{2}{\left(A_{0},A_{1},B,B_{1},H,t_{0},t_{1},V\right)}^{s}.$$Enc${}_{2,\mathrm{cca}}$$\left(PP,C_{\mathrm{cca}},SK_{1},A\right)\to C_{\mathrm{cca}}^{*}$: The broker first verifies the following equations:
$$e\left(B,g_{1}^{b}\right){=}^{?}e\left(B_{1},g_{1}\right),$$
$$e\left(V^{\prime },B_{1}\right){=}^{?}e\left(V_{1},g_{1}^{b}\right),$$
where
$$V^{\prime }=H_{2}\left(A_{0},A_{1},B,B_{1},H,t_{0},t_{1},V\right).$$
If the above two equations do not hold, the algorithm simply drops the $C_{\mathrm{cca}}$ and returns. Otherwise, the algorithm runs the original Enc${}_{1}$ algorithm and gets $C^{*}$. Then the algorithm calculates $B_{2}=g_{1}^{bs^{*}}$ and the output ciphertext will be as follows:
$$\begin{array}{cc} C^{*}=\{ & A_{0}^{*},A_{1}^{*},B,B^{*},B_{2},\left(C_{1},D_{1}\right),\ldots ,\left(C_{l},D_{l}\right),\\ & H,t_{0},t_{1},V,V_{2}\}, \end{array}$$
where
$$V_{1}=H_{2}{\left(\begin{array}{c} A_{0}^{*},A_{1}^{*},B,B^{*},B_{2},\\ \left(C_{1},D_{1}\right),\ldots ,\left(C_{l},D_{l}\right),\\ H,t_{0},t_{1},V \end{array}\right)}^{s}.$$Dec${}_{\mathrm{cca}}\left(PP,SK_{2},C_{\mathrm{cca}}^{*}\right)\to \left\{M,⊥\right\}$: The CSP first verifies the following two equations:
$$e\left(B^{*},g_{1}^{b}\right){=}^{?}e\left(B_{2},g_{1}\right),$$
$$e\left(V^{\prime },B_{2}\right){=}^{?}e\left(V_{2},g_{1}^{b}\right),$$
where
$$V^{\prime }=H_{2}\left(\begin{array}{c} A_{0}^{*},A_{1}^{*},B,B^{*},B_{2},\\ \left(C_{1},D_{1}\right),\ldots ,\left(C_{l},D_{l}\right),\\ H,t_{0},t_{1},V \end{array}\right).$$
If the above two equations do not hold, the algorithm simply returns ⊥. Otherwise, it simply runs Dec and outputs the result.

**Theorem** **4.**
*The modified scheme is CCA-secure if the original scheme is CPA-secure.*


**Proof of Theorem** **4.**The difference between the original and modified schemes is the existence of two signatures in the broker and CSP. The additional signatures have nothing to do with the confidentiality of the input messages. Since we proved the original scheme to be CPA-secure in Theorem 3, here the proof must focus only on how to answer the decryption query. Since there are two encryption processes in this scheme, we start with the Enc${}_{1}$ part. Although there is actually no entity that can decrypt the output of Enc${}_{1}$ in this scheme, we still check the CCA-security of this part. When receiving a decryption query, the oracle proceeds as follows:
If $e\left(B,g_{1}^{b}\right)=e\left(B_{1},g_{1}\right)$ and $e\left(V^{\prime },B_{1}\right)=e\left(V_{1},g_{1}^{b}\right)$ do not hold, the oracle responds ⊥.In phase 2, if the queried ciphertext is the same as the challenged ciphertext, the oracle responds ⊥.The oracle uses the master secret *s* to calculate $e{\left(g_{1},g_{1}\right)}^{\alpha s}$ and $e{\left(g_{2},g_{2}\right)}^{\beta s}$ to decrypt the queried ciphertext. The oracle then returns the decryption result to the adversary.The decryption oracle in the second encryption part is similar. So in both encryption parts, it is easy to build decryption oracles for adversaries. According to the definition in Section 3.3, the modified scheme is thus CCA-secure.  □

### 5.2. Chameleon Hash Issues

In this scheme, the sensor uses a chameleon hash to build convincing evidence for both the real and fake data. Here, we discuss three issues regarding chameleon hash functions. The first issue relates to the performance of the chameleon hash. The chameleon hash function can be treated as a trapdoor pseudo-random permutation function. That is, the cost of the chameleon hash operation is similar to that of the public key encryption operation. Some may doubt if this expensive operation is suitable for a sensor with constrained computational power. However, in the proposed scheme, the sensor does not need to run the chameleon hash function. Instead, the sensor only needs to make a random string that generates a collision, which is a lightweight operation. In our implementation, we use Krawczyk et al.’s construction [35] as the chameleon hash function. Given $m,m^{\prime },r$, the forgery string $r^{\prime }$ can be derived as follows:$$r^{\prime }=\frac{m-m^{\prime }}{x}+r.$$

So this operation is simple and affordable for a sensor. The chameleon hash function is used only in the broker and the CSP which are more powerful than the sensor.

The second issue is that one collision in a chameleon hash function implies the release of the trapdoor. In the MCCPS scenario, this would become a big problem. In many scenarios, the sensors collect data only from a limited range. For example, a thermometer sensor may only collect data from 35 ${}^{\circ }$C to 42 ${}^{\circ }$C. The attacker may try every possible combination to find the collision pair and get the trapdoor of the chameleon hash. When the trapdoor is leased, the attacker can get real data from fake data by calculating the collision. To solve this problem, in our implementation, we have added a timestamp on each piece of data to enlarge the data range and avoid the brute force attack.

The last issue to address is that, in the proposed scheme, the chameleon hash can differ on each transmission. However, this is impractical since the chameleon hash setup operation is computationally expensive. So a chameleon hash function will be used many times and it is necessary to consider the need to update the chameleon hash function. In our implementation, to ease the sensor burden, the system owner must prepare a chameleon hash function pool. When the lifetime of a chameleon hash is almost at an end, the system owner dispatches a new chameleon hash function with its corresponding trapdoor to the sensor. Note that the system owner can generate chameleon hash functions to the pool in advance. Besides, since a chameleon hash function can be independent in each transmission, there is no synchronization issue.

## 6. Conclusions and Future Works

In this paper, we proposed a privacy-preserving broker-ABE for MCCPS scenarios, in which the burden on the edge sensors is light. In this scheme, the broker is responsible for most of the encryption work. The evaluation results show the scheme’s performance is acceptable. Moreover, the data stored in clouds can be kept private even against outside coercion.

Future work will address broker security. Although in this paper, the broker is semi-trusted and cannot see the content of ciphertexts, the proposed scheme cannot ensure protection in cases where the broker and CSP are collaborating. Moreover, the broker may embed false policies and try to release data to those who are authorized to access data. So the next step is to require the broker to embed policies based on some unforgeable secrets generated by the sensors.

## Figures and Tables

**Figure 1 sensors-19-05463-f001:**
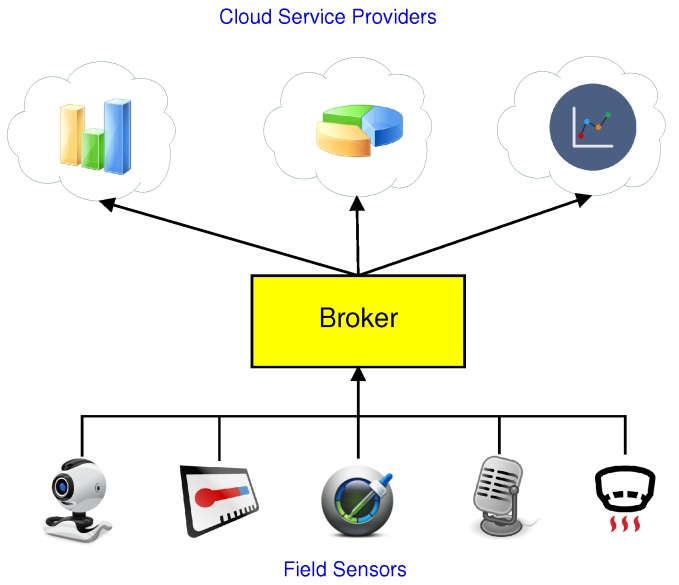
Broker architecture for the multiple cyber–physical cloud computing (MCCPS).

**Figure 2 sensors-19-05463-f002:**
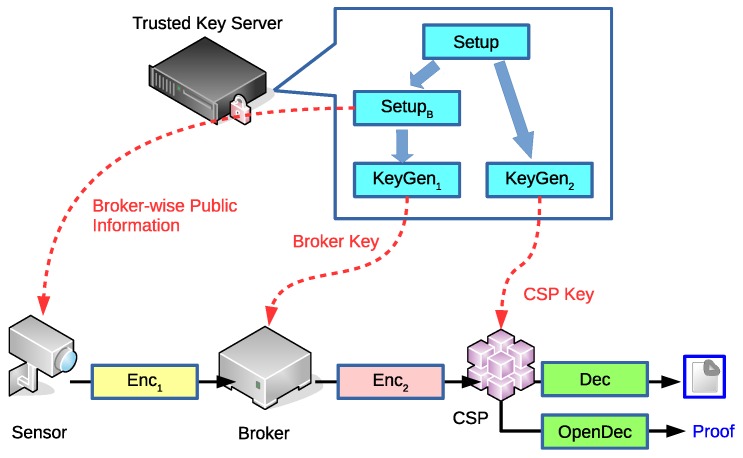
Key server, sensor, broker and cloud service provider (CSP) in the MCCPS.

**Figure 3 sensors-19-05463-f003:**
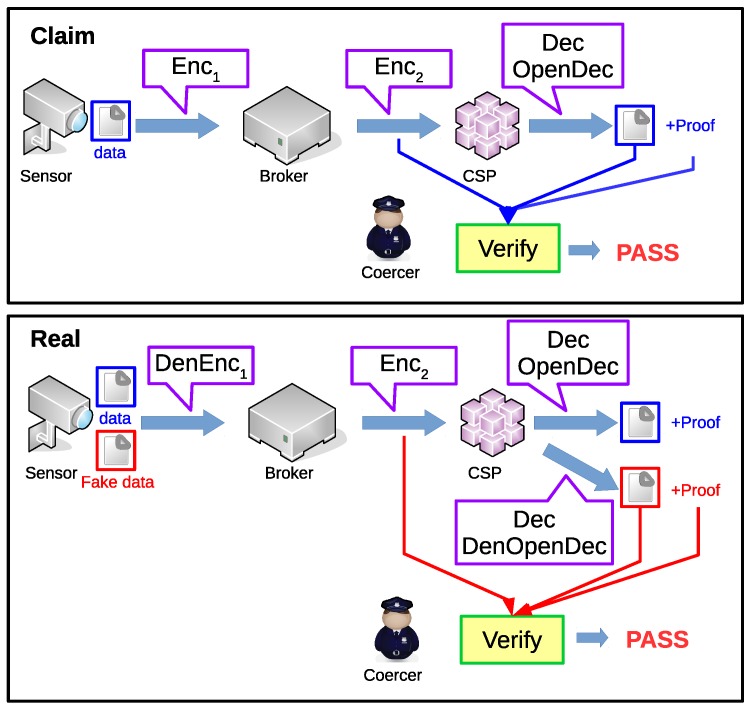
Data privacy is protected from outside coercion. The CSP can determine which algorithm should be used when being coerced. With different algorithms, the coercer will get convincing proofs for different messages.

**Figure 4 sensors-19-05463-f004:**
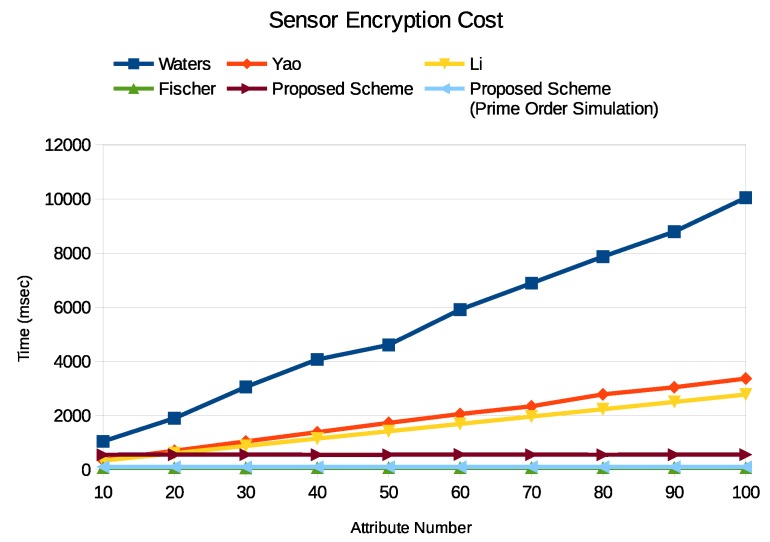
Enc${}_{1}$ execution time on Raspberry Pi3 which plays the sensor role.

**Figure 5 sensors-19-05463-f005:**
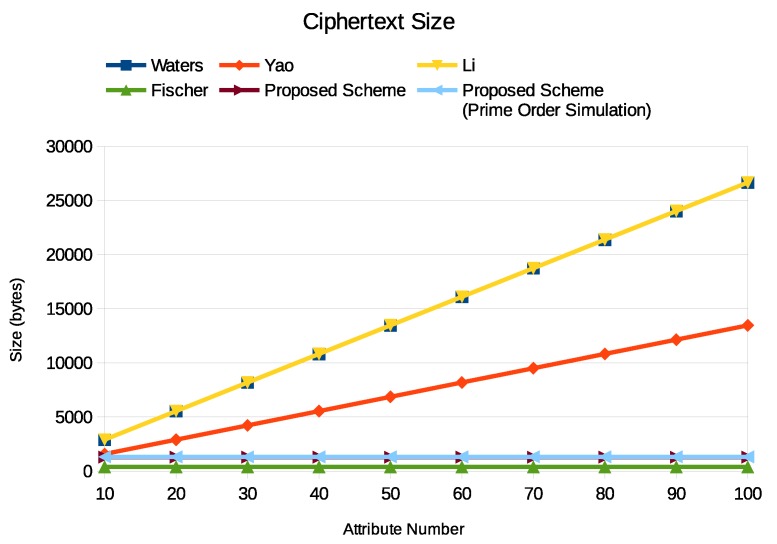
The ciphertext size of Enc${}_{1}$.

**Figure 6 sensors-19-05463-f006:**
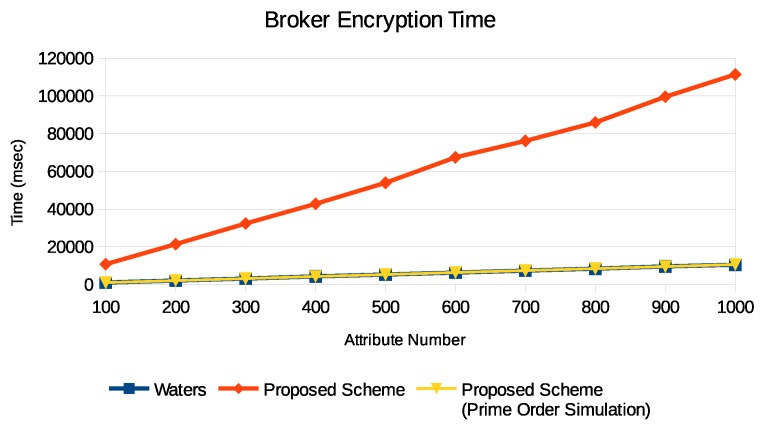
Enc${}_{2}$ execution time on a PC which plays the broker role.

**Table 1 sensors-19-05463-t001:** List of notations.

Symbol	Meaning
PP,MSK	System-wise public information and master key.
$PP_{B},MSK_{B}$	Broker-wise public information and master key.
$SK_{1},SK_{2}$	The broker’s secret key and the CSP’s secret key.
$C,C^{*},C^{\prime }$	The ciphertext encrypted by the sensor, the ciphertext re-encrypted by the broker and the ciphertext deniably encrypted by the sensor.
A,S	The access structure for a ciphertext and an attribute set.
$M,M^{\prime }$	The message and the predefined fake message.
PK	System-wise public information but not known to outsiders. That is, PK is claimed to be not existent.

**Table 2 sensors-19-05463-t002:** List all entities with their own algorithms.

Role	Algorithms
Trusted Key Server	Setup, Setup${}_{B}$, KeyGen${}_{1}$, KeyGen${}_{2}$, DenSetup, DenKeyGen${}_{2}$
CSP	Dec, OpenDec, DenOpenDec
Broker	Enc${}_{2}$
Sensor	Enc${}_{1}$, DenEnc${}_{1}$

**Table 3 sensors-19-05463-t003:** Pairing operation time.

Order	Time (ms)	Simulation Time (ms)
$p_{1}$	69.128	-
$p_{1}p_{2}$	418.222	148.912
$p_{1}p_{2}p_{3}$	1321.734	222.367
$p_{1}p_{2}p_{3}p_{4}$	2999.380	296.672
$p_{1}p_{2}p_{3}p_{4}p_{5}$	5676.130	373.077

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
