# Peer review of "Privacy-Preserving Broker-ABE Scheme for Multiple Cloud-Assisted Cyber Physical Systems"

_sensors, 2019, doi:10.3390/s19245463_

Round 1
Reviewer 1 Report
This paper proposed a privacy-preserving broker-ABE scheme for multiple cloud-assisted cyber physical systems (MCCPS), specifically for securing data communication between sensors and clouds. To ease the computational burden of the sensors, the proposed broker-ABE scheme left the policy embedding task to the broker. I think the research topic is practical. But the paper is not well organized and hard to read. And the proposed scheme may be not acceptable. Here are some major disadvantages, as follows:
1. The introduction is redundant. In addition, the authors should present the challenges of their work in Introduction part. And it was not explained who could act as the broker role in reality. 2. The proposed broker-ABE scheme aimed at decreasing the ABE computation burden at the sensors by introducing a broker entity. But there have been many lightweight ABE works proposed to achieve the same property. I think the authors should add the discussion about the existing lightweight ABE works into the Sections of Introduction and Related works. Moreover, the proposed broker-ABE scheme should be compared with the existing lightweight ABE works. 3. In Introduction, the authors claimed that the traditional encryption schemes would not protect user privacy since CSPs hand unencrypted data. But the proposed broker-ABE scheme still let the CSP to execute decryption to obtain the unencrypted data. So, I feel that the proposed broker-ABE scheme didn’t solve the privacy leakage problem presented in Introduction. The proposed scheme is probably over-estimated. 4. I think the security of the system model is not sufficiently practical. In reality, the cloud is always untrusted. The cloud could leak the confidential information intentionally or accidentally. The authors should take such an untrusted cloud scenario into consideration. 5. To make the paper easy-to-read, the authors are suggested to add a Section to explain the system model, especially the security levels of the four roles, i.e., a trusted key server, a CSP, a broker and a sensor. 6. In Section 4.1, the algorithm $DenKeyGen_2$ is not consistent with the corresponding description. 7. Observing the algorithm $DenEnc_1$, it seems that a fake message $M’$ is chosen and encrypted for each real message $M$. Is the total encryption overhead doubled? 8. On page 10, below the 12th algorithm, the authors said there was no $DenEnc_2$, $DenKeyGen_1$, $OpenEnc$, and $DenOpenEnc$. But why do these algorithms appear? I feel the scheme is in chaos, hard to understand, and seems self-contradictory. A flow diagram should be added to describe the protocol. 9. The performance evaluation is not enough. First, it lacks the comparison between the proposed scheme with the latest and related lightweight ABE. It is not sufficient to judge whether the proposed scheme is acceptable, when taking just the Waters CP-ABE scheme in 2011 as comparison reference. Second, since the proposed scheme is used for securing data communication, I think that the communication data size, i.e., the message ciphertext, and communication delay should be evaluated as well. In one word, the authors should provide an overall evaluation. 10. Most of the references are not new enough. Some latest and high-quality references need to be discussed.

Reviewer 2 Report
The paper proposes a privacy-preserving broker-ABE scheme for cloud-assisted CPS.
Various existing work is incorporated to make the proposed scheme work. Having said that, this makes the research contribution of the paper difficult to distinguish. Section 3 is largely previous work. Possibly fit better within section 2, or add something like "preliminary" in the section title.
Section 4 brings those ideas together as a process. What could be more distinctive are:
1) construct the framework, extending the process developed here.
2) just before section 4.2., define a set of threats and construct an attack model.
3) Figure 2: should be redrawn for the framework (the current version is not very helpful or intuitive). Also, the process can be captured better in details. Then, such processes can be referenced in the body for a better understanding of its features.
Hard to fault the setup as they are built on previous work. Instead, authors should do better capturing the use of multiple techniques better (hence, the framework).
Going back to point (2), it would be good to see the coverage of how the proposed method can mitigate some critical attacks. Hence, the attack model would be a good idea.
Round 2
Reviewer 1 Report
This paper proposed a privacy-preserving broker-ABE scheme for multiple cloud-assisted cyber physical systems (MCCPS). The authors claimed that it could protect the data from the outside but powerful entity, not from the CSP. But In this case, I still have some problems and suggestions, as follows:
In Introduction, the authors claimed that the traditional encryption schemes would not protect user privacy because CSPs handle unencrypted data, while the proposed broker-ABE scheme still let the CSP to execute decryption to obtain the unencrypted data and claimed that it was secure. The authors explained that their scheme was under a secure CSP. But I feel that the traditional encryption schemes would also protect user privacy under the same security environment. Thus, if the traditional encryption schemes have a different security environment from that in this work, or even a weaker secure one, there is no comparability between this work and the traditional schemes, in the security regard. Moreover, the proposed scheme might be not securer than the traditional encryption scheme. Since the authors studies under secure CSP, the existing works with the same secure environment should be compared. Otherwise, it is not fair to “the traditional encryption schemes”. As for the previous comment “To make the paper easy-to-read, the authors are suggested to add an Section to explain the system model, especially the security levels of the four roles, i.e., a trusted key server, a CSP, a broker and a sensor.”, the authors said in Section 4.3, the security levels were described. First, I didn’t find the description in Section 4.3. Second, there should be a separate section to explain the system model, to make this paper easy to read. For example, which role is trusted or untrusted, what kinds of attacks may happen, etc. Third, since all the four roles are trusted, the authors should illustrate which role the proposed scheme tried to protect from. And this role should be shown in Figure 3. The authors should highlight the challenges, by even spending one paragraph. Especially, please explain whether it is hard to realize deniability, and why.

Reviewer 2 Report
Issues have been addressed well in the revised version.
Author Response
Thanks for your comments to make our work better.
Round 3
Reviewer 1 Report
The proposed broker-ABE scheme was claimed to keep secure, by two steps: first generating a fake key to encrypt a fake message; second, leaking just the fake key to the coercion.
In this respect, I have two questions:
Since, the cloud needs to first judge whether an access query request is from the coercion or not. If so, the cloud will release the fake key and the corresponding fake message's ciphertext to the coercion. But if the coercion gets the true message's ciphertext (which is easy since the ciphertext is usually public) and finds that he cannot decrypt by the fake key. Finally, the fake key "lie" to the coercion will be exposed. "first generating a fake key to encrypt a fake message; second, leaking just the fake key to the coercion" this two-steps mechanism is also easy to think up and realize in other ABE schemes. Please present clearly the technical challenges of this work.
